# RUSID: Robust Uncertainty-aware Single Image Deraining beyond Certainty

## Abstract

Rainy weather induces rain streaks, blurs details, and reduces contrast, impairing image quality, making single image deraining a classic research topic. However, existing learning-based image restoration methods fail to account for uncertainties in both data and model dimensions, thus being unable to produce satisfactory results. To address this challenge, we introduce a novel framework called the Uncertainty-aware Visual-priors Prompt-interaction Network (UVPNet). UVPNet comprises three key modules: the Distribution-aware Visual Priors Learning (DVPL) module, which aims at data-wise aleatoric uncertainties, the Certainty-Uncertainty Prompt Fusion (CUPF) module, which tackles model-wise epistemic uncertainties, and the Channel Spatial Uncertainty Weighting Block (CSUWB). UVPNet leverages uncertainty modeling through visual semantic and depth priors and distributionally representative prompts by integrating data-wise and model-wise uncertainty learning. To the best of our knowledge, our UVPNet first utilizes uncertainty modeling with visual priors for single image deraining. Extensive experiment results demonstrate that our UVPNet outperforms state-of-the-art methods on both public synthetic datasets and real-world images while maintaining low complexity.

## 1 Introduction

Adverse weather conditions present formidable obstacles to visual perception tasks across a wide spectrum of applications such as Autonomous Driving(AD). In the domain of AD, rain can cause raindrops to obscure the camera lens, leading to blurred and distorted images of the road, traffic signs, and other vehicles. This can severely impede the vehicle's ability to accurately detect and classify objects, potentially resulting in dangerous driving situations. Fog, on the other hand, reduces visibility by scattering light, making it difficult for cameras to distinguish between different objects in the scene. In surveillance systems, fog can render the captured images nearly useless for identifying individuals or detecting suspicious activities. Rain, prevalent in arid and semi-arid regions, not only introduces a large amount of particulate matter into the air but also causes light attenuation and scattering. In outdoor photography, rainy weather can turn a clear and beautiful landscape into a hazy and indistinguishable scene. These challenging weather conditions have spurred extensive research in the field of image restoration. A multitude of studies, as referenced inJiang et al. (2020); Zamir et al. (2021; 2022), have been dedicated to this area. The overarching goal of this research is to enhance the clarity and quality of images under adverse weather conditions. By improving image quality, more reliable visual data can be provided for various applicationsValanarasu et al. (2022), such as surveillanceZheng et al. (2021b); Cui & Knoll (2023) and remote sensingHe et al. (2010); Song et al. (2023). In autonomous driving, high-quality images enable the vehicle's perception system to make more informed decisions, ensuring safer driving.

However, the complex nature of rainy weather poses unique challenges for current image restoration methods. Current methods cannot solve the high uncertainty problem. Rain streaks and rain drops introduce multiple degradation types simultaneously. These models typically focus on a single aspect of degradation, such as dehazing or denoising, and struggle to capture a comprehensive set of features when dealing with the complex interplay of multiple degradations in rain-affected imagesZamir et al. (2021). This uncertainty makes it challenging for these models to accurately estimate the true nature of the degradation and restore the image to its original state.

The inherent uncertainties associated with these multiple degradation types are difficult for existing methods to handle effectively. The high uncertainty of rain particles in the air can cause light scattering, absorption, and occlusion. As a significant factor in rainy image degradations, uncertainty-aware probabilistic modeling becomes even more critical.

In response to challenges of dealing with the high uncertainty of rainy images, we propose the Uncertainty-aware Visual-priors Prompt-interaction Network (UVPNet) in this paper. UVPNet consists of our proposed Distribution-aware Visual Priors Learning(DVPL) module and Certainty-Uncertainty Prompt Fusion(CUPF) module, which can represent aleatoric and epistemic uncertainty, respectively. Specifically, for data-wise uncertainty, our proposed DVPL models the visual priors of segment and depth under uncertainty, enabling the model to extract uncertainty information to guide its training. For model-wise uncertainty, our proposed CUPF allows prompt learning to capture the uncertainty with the fusion of a certainty prompt, thereby enabling the decoder to probabilistically reconstruct clear images. Furthermore, to address the uncertainty inherent in the neural network while enhancing the quality of restored images, we propose the Channel Spatial Uncertainty Weighting Block(CSUWB). CSUWB enhances UVPNet's performance through uncertainty-aware feature processing.

In summary, the primary contributions of this paper can be categorized into four main aspects:

- To the best of our knowledge, we are the first to utilize uncertainty modeling with the segment and depth priors for the single image deraining task.

- We propose the Certainty-Uncertainty Prompt Fusion module to handle epistemic uncertainty in model dimensions, effectively enhancing model robustness.

- We propose the Channel Spatial Uncertainty Weighting Block to guide the feature selection process of our proposed UVPNet, thereby boosting overall performance.

- Through comprehensive experiments conducted on several public deraining datasets and real-world images, we demonstrate that our proposed method achieves **S**tate-**O**f-**T**he-**A**rt(**SOTA**) performance.

## 2 METHODS

### 2.1 OVERALL FRAMEWORK

We propose UVPNet in this paper to address the lack of research on high uncertainty in the task of single-image rain removal. The overall structure of our proposed UVPNet is as depicted in Fig. 1. Our network adopts the multi-input multi-output U-shaped encoder-decoder structure due to its excellent performance. The encoder receives three inputs, and the decoder outputs restored images at three different scales. We use four NAFBlocksChen et al. (2022) as the backbone network embedded in each CSUWB. At the encoder end, we also utilize the prior information provided by SAMKirillov et al. (2023) and Depth Anything V2Yang et al. (2024) in an uncertainty-aware manner; meanwhile, at the decoder end, we also employ prompt learning modules with uncertainty-aware capabilities.

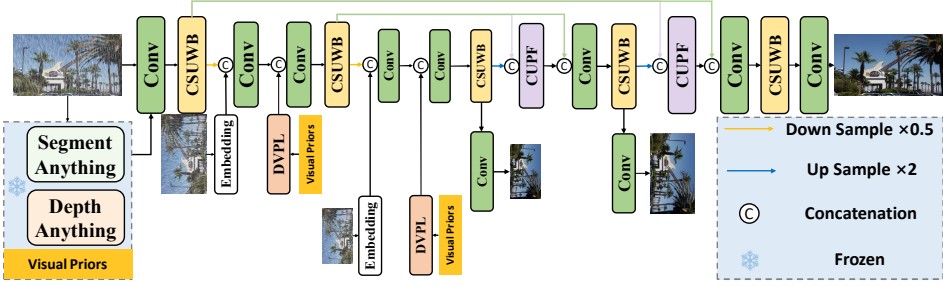

Figure 1: Overall framework of our proposed UVPNet.

## 2.2 Distribution-aware Visual Priors Learning

To address the uncertainty in data dimensions for single image rain removal, we integrate uncertainty-aware visual priors provided by the Distribution-aware Visual Priors Learning(DVPL) into our UVPNet, thereby achieving an accurate representation for aleatoric uncertainties. We utilize a learnable distribution in the feature space to introduce uncertainty modeling into the rain streak and rain drop removal task. We use two independent convolutional layers $f_\mu$ and $f_\sigma$ to achieve the feature distribution, and we model the distribution of DVPL at each pixel as Gaussian following Yang et al. (2021). The results are denoted as the mean $\mu$ and the variance $\sigma$ of these two layers. These two convolutional layers learn a distribution $N(\mu, \sigma)$ parameterized by mean $\mu$ and variance $\sigma$. This approach allows the model to focus on the most valuable aspects inside the features while being aware of the inherent high uncertainty region in the feature space. The detailed formulation of this uncertainty-aware distribution modeling can be shown as:

$$Y \sim N(f_u(x), \epsilon \odot \sigma(x)) \tag{1}$$

where

$$\sigma(x) = Softplus(f_\sigma(x)) + \theta \tag{2}$$

where $x$ and $Y$ are the input feature and output result, $Softplus$ denotes the Softplus activation function, $\theta$ is set to 0.001, $\odot$ denotes the element-wise multiplication, and $\epsilon \sim N(0, 1)$. As shown in Fig.2(b), we apply the aforementioned distribution modeling process to extract the uncertainty within the segment and depth priors provided by SAMKirillov et al. (2023) and Depth Anything V2Yang et al. (2024), using it to characterize the data-wise aleatoric uncertainty. Furthermore, we employ the affine transformation to refine the features, leveraging this approach to enhance the representation of the data-wise inherent uncertainty. Finally, we couple these selected features with the original feature. This process can be described by the following *Eq.* 3:

$$\begin{aligned} P_S &= Conv([S_w * x + S_b, x]), \\ P_D &= Conv([D_w * x + D_b, x]) \end{aligned} \tag{3}$$

where $S_w$, $S_b$, $D_w$, and $D_b$ are obtained by modeling the uncertainty of the segment priors and depth priors in an affine transformation way, $P_S$ and $P_D$ are the learned priors. $[,]$ indicates the concatenation operation. Further, we designed a cross-branch weighted interaction to facilitate the cross-interaction between segmentation priors and depth features, as shown in the right part of Fig. 2(a). The goal is to cross-complement one type of visual prior feature with another. Segment prior features include details of edges and fine texture categories, so we utilize this information to enrich the exploration of depth visual prior features through a lightweight spatial feature weighting unit (Segment to Depth, S→D), as shown in Fig. 2(c). Similarly, the global information in depth prior features is passed to the segmentation prior uncertainty-aware distribution learning branch through a channel feature weighting unit (Depth to Segment, D→S), as shown in Fig. 2(d).

**S→D:** This module calculates the weight map of the spatial dimension based on the prior features of semantic segmentation, and then uses it to supplement the features of the depth prior branch, thereby improving the overall model feature extraction ability. The S→D unit utilizes the average pooling and max pooling of channel dimensions in parallel to generate two independent spatial feature maps, each with a size of $H \times W \times 1$. Then, these two feature maps are concatenated in the channel dimension, and the concatenated features are further refined through a convolution layer with a kernel size of $7 \times 7$. Finally, the sigmoid activation function is used to generate the final spatial attention map. Overall, the detailed process of the S→D module can be formulated as:

$$A_{S \to D} = \delta(f_{7 \times 7}^{2 \to 1}([GAP_c(x), GMP_c(x)])) \tag{4}$$

where $f_{7 \times 7}$ is the convolution layer with a kernel size of $7 \times 7$ and changes the channel of input features from 2 to 1; $\delta$ is the sigmoid function, $GAP_c$ and $GMP_c$ are the channel-wise global average pooling and max pooling, respectively.

**D→S:** D→S is an approximately symmetrical dual-branch module that processes the input depth visual prior features, generates a weight, and then uses it to process the segmentation prior features of uncertainty perception. Specifically, given the input depth prior feature $x \in \mathbb{R}^{H \times W \times C}$, the first branch of the D→S module applies global average pooling along the spatial dimension to obtain feature vectors of size $1 \times 1 \times C$, followed by two convolutional layers with an intermediate ReLU

activation. The other branch of the D→S module adopts the same structure, with the only difference being the use of max pooling to obtain features. Finally, add the results of the two branches and apply the sigmoid function to generate the final weight, which is used to modulate the uncertainty-aware segmentation prior. The detailed process of the D→S module can be expressed as:

$$
\begin{aligned}
A_{D \to S} = \delta(&f_{1 \times 1}^{\frac{c}{r} \to c}(\gamma(f_{1 \times 1}^{c \to \frac{c}{r}}(GAP_s(x)))) \\
&+ f_{1 \times 1}^{\frac{c}{r} \to c}(\gamma(f_{1 \times 1}^{c \to \frac{c}{r}}(GMP_s(x)))))
\end{aligned}
\tag{5}
$$

where $GAP_s$ and $GMP_s$ are the spatial-wise global average pooling and max pooling, respectively; $\gamma$ denotes the ReLU activation function. $f_{1 \times 1}^{c \to \frac{c}{r}}$ has a reduction ratio $r$ for the channel adjustment, while $f_{1 \times 1}^{\frac{c}{r} \to c}$ has an increasing ratio $r$.

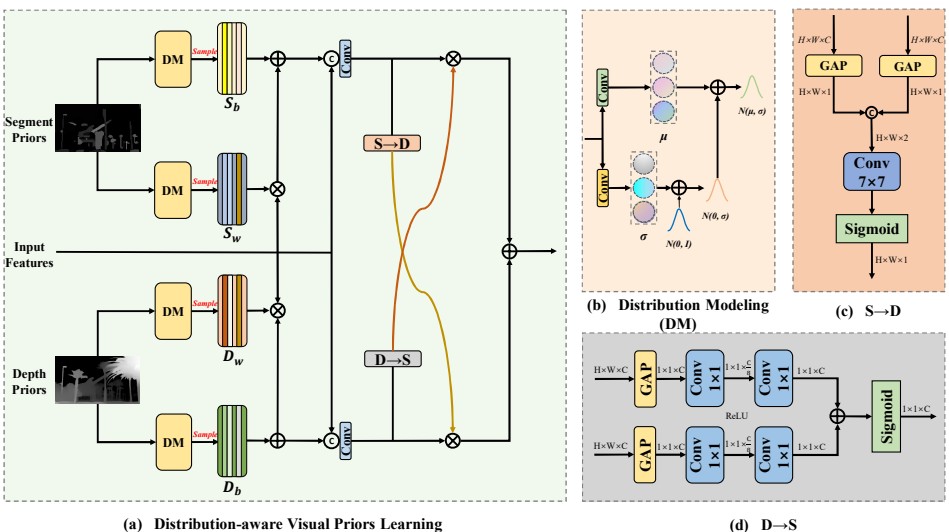

Figure 2: Detailed structure of our proposed DVPL.

## 2.3 CERTAINTY-UNCERTAINTY PROMPT FUSION

We introduce the Certainty-Uncertainty Prompt Fusion (CUPF) module engineered to mitigate the epistemic uncertainty intrinsic to neural networks. The inherent uncertainty within a model is termed epistemic uncertainty, and neural network models designed to learn fixed mappings lack the capability to capture this uncertainty in the model dimension. To address this issue, CUPF integrates prompt learning with uncertainty-aware distribution modeling, enabling the model to represent its own inherent epistemic uncertainties with the fusion of certainty prompts.

Prompt-based techniques have been investigated for the parameter-efficient fine-tuning of large, frozen models trained on a source task to adapt them to a target task within the domains of NLPBrown et al. (2020); Houlsby et al. (2019) and visionJia et al. (2022); Khattak et al. (2023). However, prompt learning lacks the capability to address uncertainties in the model dimensions. Consequently, we integrate prompt learning with uncertainty-aware distributional representations, thereby transforming prompt learning into a mitigator of model dimension epistemic uncertainties. In our proposed CUPF module, the prompt components serve as learnable parameters that interact with input features, thereby enriching the input features with information regarding the degree of rainy degradation. Given input features $x \in \mathbb{R}^{H \times W \times C}$ and $N$ certainty prompt components $P_{cer} \in \mathbb{R}^{N \times H \times W \times C}$, and $N$ uncertainty prompt components $P_{unc} \in \mathbb{R}^{N \times H \times W \times C}$, the distribution-aware prompt components can be obtained by:

$$
P_{dis} \sim P_{cer} + \epsilon \odot \sigma_p(P_{unc})
\tag{6}
$$

where

$$
\sigma_p(P_{unc}) = Softplus(P_{unc}) + \theta
\tag{7}
$$

then the final fusion prompt components $P$ are the sum of $W_c \odot P_{cer}$ and $W_d \odot P_{dis}$, where $W_c$ and $W_d$ are learnable parameters used for weighting this two branch. So the detailed formulation of the output $Y_{CUPF}$ of our CUPF can be defined as:

$$Y_{CUPF} = PIM(PGM(P, x), x) \tag{8}$$

where $PGM$ and $PIM$ are the prompt generation module and prompt interaction modulePotlapalli et al. (2023), respectively.

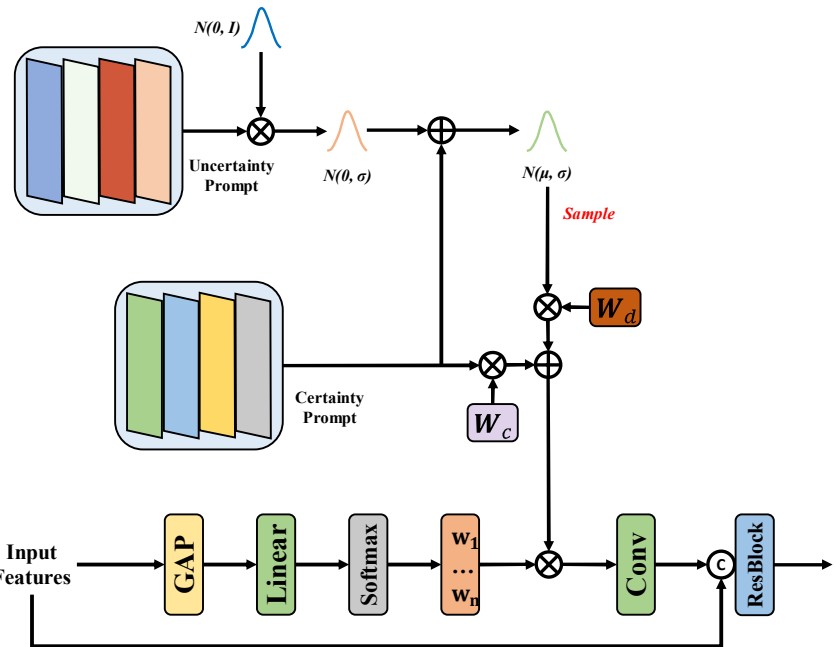

Figure 3: Detailed structure of our proposed CUPF.

## 2.4 CHANNEL SPATIAL UNCERTAINTY WEIGHTING BLOCK

This section introduces a novel feature representation method proposed in this paper, called Channel Spatial Uncertainty Weighting Block (CSUWB). In common standard convolutional networks, channel dimension mapping is performed on input features. There are many studies that weigh and refine the features among channels to improve the feature representation ability of neural networks. However, the weights corresponding to each channel in a convolutional neural network are randomly initialized, and the correspondence between multiple channels is also random, which results in significant uncertainty in the channel dimension of the convolutional neural network. To address this issue and characterize the uncertainty of channel dimension changes in convolutional neural networks, we propose Channel-wise Uncertainty-aware Weighting Block (CUWB), as shown in Fig. 4(a) and (b). Specifically, given input features $x$, we first project the input features through convolutional layers, while also using convolutional layers to project the features that have undergone channel shuffle operations; Then, the shuffled features are sent to an unshuffle operation, and the two projected features are subtracted element by element to obtain the uncertainty features $U_c$ of the channel dimension, this process can be formulated as:

$$U_c = |Conv(x) - UShu(Conv(Shu(x)))| \tag{9}$$

where $Conv$ denotes the convolution layer for projection, $UShu$ and $Shu$ are the channel-wise unshuffle operation and shuffle operation, respectively. Then, the obtained uncertainty features are subjected to global average pooling and global maximum pooling in spatial dimensions, and the feature information is aggregated into channel statistics. These statistics are input into an MLP with a reduction rate to generate channel attention weights. This process can be formulated as follows:

$$W_c = \delta(MLP(GAP_s(U_c) + GMP_s(U_c))) \tag{10}$$

These weights are used to scale the original features, highlighting channels related to uncertainty perception while suppressing irrelevant channels. Finally, the weighted uncertainty-aware features are concatenated with the original deterministic features, and the final mapping is performed by a convolution layer.

Similar to CUWB, we propose a novel module for measuring the uncertainty caused by feature changes in spatial dimensions, called the Spatial-wise Uncertainty Weighting Module (SUWM). Most neural networks used for image restoration are based on UNet architecture or Encoder-Decoder architecture, which inevitably require frequent sampling operations of features to reduce computational complexity. Although many studies have investigated how to weigh spatial features, these studies have overlooked the impact of uncertainty caused by changes in feature maps on model performance. Meanwhile, since the model is generally trained using a fixed image size, the actual application of the model involves significant changes in image size, which introduces considerable uncertainty. So it is necessary to model the uncertainty caused by size changes in the spatial dimension of features. To address this issue and characterize the uncertainty of spatial dimension changes in UNet or Encoder-Decoder architecture, we propose our novel SUWM in this section, as shown in Fig. 4(c) and (d). Unlike CUWB, we use convolutional layers to project both the original input features and the downsampled spatially transformed features, specifically. Then, the downsampled projected features are upsampled to restore their original spatial size, and then subtracted element by element from the original features to obtain uncertain spatial features $U_s$. This process can be formulated as:

$$U_s = |Conv(x) - Up(Conv(Down(x)))| \tag{11}$$

where $Up$ and $Down$ are upsampling operation and downsampling operation, respectively. Then, the obtained uncertainty features are characterized using global average pooling and global maximum pooling in the channel dimension, and are input into a convolutional layer for refinement. Finally, the spatial uncertainty weights are obtained through a sigmoid layer. This process can be formulated as:

$$W_s = \delta(Conv[GAP_c(U_s), GMP_c(U_s)]) \tag{12}$$

Finally, we fuse the features refined by CUWB and SUWM with the original features $x$ represented by 4 NAFBlocksChen et al. (2022) using a residual connection. This process can be expressed as follows:

$$Y_{CSUWB} = x + x * W_c * W_s \tag{13}$$

We use CSUWB to weight the uncertainty perception of the original features. After refinement by CSUWB, the original features have the ability to sense uncertainty, greatly improving the performance of single image rain removal.

## 3 EXPERIMENTS

### 3.1 IMPLEMENTATION DETAILS AND DATASETS

We implement our framework and other methods using PyTorch. We train all the models for 200 epochs with the AdamW optimizer from scratch for fairness. The initial learning rate is set to $1 \times 10^{-4}$, and we employ CosineAnnealingLR to adjust the learning rate. For data augmentation, we apply horizontal flipping and randomly rotate the image by 0, 90, 180, 270 degrees. The batch size is set to 4. Computing complexity is computed on a patch size of $128 \times 128$.

### 3.2 RESULTS

For datasets used for training and testing on the image rain streak removal task, we use 13,712 image pairs collected from multiple datasetsLi et al. (2016); Yang et al. (2017); Zhang et al. (2019); Zhang & Patel (2018); Fu et al. (2017) for training by following previous methodsZamir et al. (2021; 2022); Cui et al. (2023). For testing the performance of rain streak removal, we use five synthetic datasets, Test100Zhang et al. (2019), Rain100LYang et al. (2017), Rain100HYang et al. (2017), Test1200Zhang & Patel (2018), Test2800Fu et al. (2017), and a public real-world rainy image datasetWang et al. (2019). Furthermore, for evaluating the performance of rain drop removal, we use the public datasetQian et al. (2018) for training and testing. In the comparative experiments with other methods, we compare our proposed method with various cutting-edge approaches and compute PSNR and SSIM scores following Li et al. (2025). For real-world rainy images, we use

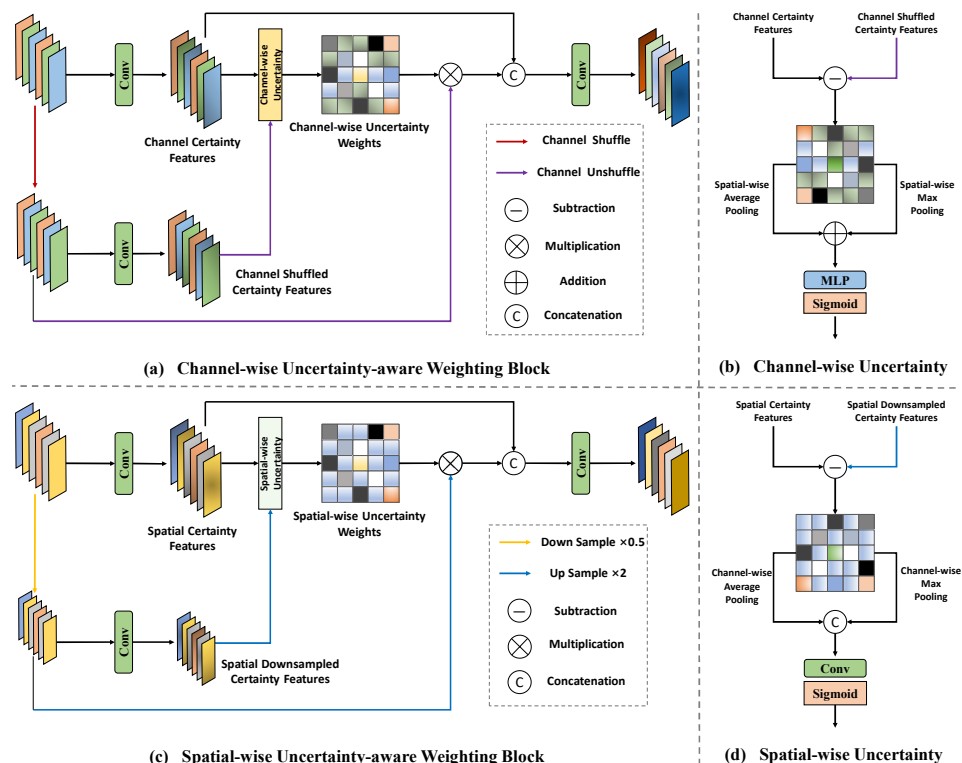

Figure 4: Detailed structure of our proposed CSUWB.

NIQEMittal et al. (2012b), PIQEVenkatanath et al. (2015), and BRISQUEMittal et al. (2012a) scores for evaluation. For the MaIRLi et al. (2025) method, we utilize the UNet variant model specifically designed for image dehazing in their proposed approach.

For the rain streak removal task and rain drop removal task on synthetic images, as shown in Table 1 and Table 3, our method achieves significant performance gains more than other methods while maintaining fewer parameters and computational complexity as shown in Table 1. Red/blue text indicates best/second-best comparative results, the third-best comparative results are underlined. Furthermore, the visual results shown in Fig. 5 are in close alignment with the quantitative findings, substantiating the superior image rain streak removal capabilities of our proposed method. Various methods produced relatively similar restored images on synthetic data, making it difficult to distinguish differences visually. So, we use error mapsZheng et al. (2021a) between restoration results and ground-truth clear images to demonstrate the effectiveness of each method in restoring images. Our method produces the clearest error map, while others like MaIRLi et al. (2025), MFSNetGao & Dang (2025), and PromptIRPotlapalli et al. (2023) show many errors in regions with sparser features, such as buildings and skies. Other methods, such as FMRNetJiang et al. (2024), MHNetGao et al. (2025), and CPRAFormerZou et al. (2025), perform poorly in the rain removal task, thus exhibiting larger errors in the error maps.

For real-world rainy images, as shown in the bottom two rows of Fig. 5, except for our method, which effectively removes rainy streaks from images, all other methods fail to do so in real-world scenarios. Additionally, no reference image quality metrics shown in Table 2 denote that our method can removal real-world rain streaks effectively.

### 3.3 DISCUSSION ON COMPUTING COMPLEXITY

In this part, we have listed relevant complexity measurement metrics, including Inference times (in ms), and Memory cost (in Mb), measured using a NVIDIA RTX4090 GPU. As shown in Table 1, our method maintains low Flops and parameters while achieving SOTA performance, while only occupying low memory at runtime and achieving fast inference time. Other methods, such as

MIMOUNet, NAFNet, and FMRNet, although simpler than our proposed method in a certain complexity metric, have poor image rain removal performance and cannot meet the quality requirements of actual image restoration. Some methods, such as PromptIR, MHNet, and MFSNet, significantly exceed the proposed method in terms of computational complexity metrics, and have shortcomings in image rain removal performance, making them cumbersome. Finally, the MaIR method is based on Mamba, and due to the complex image scanning operations involved, although it can maintain low Flops and Params, it consumes the most inference time.

Furthermore, due to the use of vision large models (i.e., SAM and Depth Anything V2) in our UVPNet to obtain priors, we also conducted research on scenarios where vision large models cannot be used. As shown in Table 5. The 0S, 0D, wS, and wD represent segmentation priors with tensor values all 0, depth priors with tensor values all 0, normal SAM priors, and Depth Anything V2 priors, respectively. From Table 5, it can be seen that without relying on SAM and Depth Anything V2, that is, when inputting all 0 tensors, although our method has a performance decrease, it is not significant and does not affect SOTA performance. This indicates that our method has the potential for real-time applications and can handle real-time requirements in application scenarios.

## 3.4 ABLATION STUDIES

For simplicity, we randomly selected 4,000 images from all the previous dataset used for training. As shown in Table 4, the ablation study of the above-mentioned modules proves the effectiveness of our proposed method. For example, using segment priors and depth priors in DVPL can improve performance. Meanwhile, fusing the certainty prompt and uncertainty prompt in the CUPF module can significantly enhance network performance. Finally, by embedding CUWB and SUWM in CSUWB can truly improve the image quality than using CBAMWoo et al. (2018) or Transformer blockZamir et al. (2022).

Table 1: Comparative results on Raindrop removal dataset.

| Method | Venue | Raindrop-A | | Raindrop-B | | Inf. times | Mem. cost |
|---|---|---|---|---|---|---|---|
| | | PSNR↑ | SSIM↑ | PSNR↑ | SSIM↑ | ms | Mb |
| MIMOUNetCho et al. (2021) | ICCV'21 | 26.70 | 0.906 | 24.52 | 0.871 | 2.260 | 25.97 |
| NAFNetChen et al. (2022) | ECCV'22 | 26.04 | 0.893 | 24.10 | 0.860 | 6.493 | 65.28 |
| PromptIRPotlapalli et al. (2023) | Nips'23 | 28.09 | 0.931 | 25.12 | 0.887 | 16.453 | 135.77 |
| FMRNetJiang et al. (2024) | AAAI'24 | 23.40 | 0.815 | 22.11 | 0.796 | 17.562 | 6.63 |
| MHNetGao et al. (2025) | PR'25 | 25.31 | 0.882 | 23.64 | 0.851 | 13.476 | 257.66 |
| MFSNetGao & Dang (2025) | TCSVT'25 | 27.01 | 0.910 | 24.62 | 0.873 | 14.601 | 351.54 |
| CPRAformerZou et al. (2025) | ACMMM'25 | 28.13 | 0.932 | 25.14 | 0.887 | 32.548 | 111.68 |
| MaIRLi et al. (2025) | CVPR'25 | 27.24 | 0.917 | 24.65 | 0.875 | 53.165 | 12.99 |
| Ours | - | 28.36 | 0.935 | 25.28 | 0.892 | 8.580 | 49.34 |

Table 2: Comparative results on real_test_1000 datasetWang et al. (2019).

| Method | PromptIR | FMRNet | MHNet | MFSNet | CPRAFormer | MaIR | Ours |
|---|---|---|---|---|---|---|---|
| NIQE↓ | 6.00 | 7.65 | 8.20 | 8.01 | 5.96 | 7.32 | 5.86 |
| BRISQUE↓ | 40.70 | 39.98 | 39.64 | 40.32 | 41.34 | 41.95 | 39.82 |
| PIQE↓ | 58.18 | 64.48 | 64.58 | 66.94 | 58.80 | 71.58 | 56.85 |

Table 3: Results of synthetic rainy images on several public datasets.

| Method | Venue | Rain100L | | Rain100H | | Test1200 | | Test100 | | Test2800 | | Average | |
|---|---|---|---|---|---|---|---|---|---|---|---|---|---|
| | | PSNR↑ | SSIM↑ | PSNR↑ | SSIM↑ | PSNR↑ | SSIM↑ | PSNR↑ | SSIM↑ | PSNR↑ | SSIM↑ | PSNR↑ | SSIM↑ |
| MIMOUnetCho et al. (2021) | ICCV'21 | 29.68 | 0.931 | 27.47 | 0.899 | 29.48 | 0.934 | 29.04 | 0.931 | 28.86 | 0.937 | 28.90 | 0.937 |
| NAFNetChen et al. (2022) | ECCV'22 | 30.73 | 0.950 | 27.62 | 0.898 | 28.66 | 0.930 | 28.41 | 0.928 | 28.50 | 0.934 | 28.78 | 0.928 |
| PromptIRPotlapalli et al. (2023) | Nips'23 | 31.08 | 0.938 | 28.01 | 0.900 | 28.42 | 0.928 | 28.15 | 0.926 | 28.35 | 0.935 | 28.80 | 0.925 |
| FMRNet Jiang et al. (2024) | AAAI'24 | 28.17 | 0.903 | 25.91 | 0.859 | 26.94 | 0.921 | 26.72 | 0.919 | 27.33 | 0.926 | 27.01 | 0.905 |
| MHNetGao et al. (2025) | PR'25 | 31.12 | 0.951 | 27.67 | 0.896 | 28.01 | 0.925 | 27.77 | 0.923 | 28.17 | 0.932 | 28.54 | 0.925 |
| MFSNetGao & Dang (2025) | TCSVT'25 | 30.68 | 0.946 | 27.89 | 0.904 | 28.01 | 0.926 | 27.76 | 0.924 | 28.12 | 0.933 | 28.49 | 0.926 |
| CPRAformerZou et al. (2025) | ACMMM'25 | 32.73 | 0.968 | 29.03 | 0.920 | 27.74 | 0.923 | 27.49 | 0.922 | 27.74 | 0.930 | 28.94 | 0.932 |
| MaIRLi et al. (2025) | CVPR'25 | 31.05 | 0.949 | 28.24 | 0.905 | 29.46 | 0.934 | 29.15 | 0.932 | 28.93 | 0.937 | 29.36 | 0.931 |
| Ours | - | 31.72 | 0.955 | 28.69 | 0.913 | 29.50 | 0.938 | 29.21 | 0.936 | 29.00 | 0.939 | 29.62 | 0.936 |

## 4 CONCLUSION

In this paper, we propose Uncertainty-aware Visual-priors Prompt-interaction Network(UVPNet) to address the impact of high-uncertainty in rainy scenes on visual perception data. For the first

Table 4: Ablation studies of components proposed on Test1200 datasetZhang & Patel (2018).

| Configuration | Experiment setting | PSNR↑ | SSIM↑ |
|---|---|---|---|
| Baseline | - | 28.34 | 0.887 |
| DVPL | w/o Segment priors | 28.56 | 0.888 |
| | w/o Depth priors | 28.60 | 0.891 |
| | w/o uncertainty modeling | 28.76 | 0.891 |
| | w/o attentive weighting | 28.62 | 0.891 |
| | Ours | 28.85 | 0.892 |
| CUPF | fixed all prompt | 28.60 | 0.890 |
| | fixed only certainty prompt | 28.63 | 0.889 |
| | w/o certainty prompt | 28.70 | 0.893 |
| | w/o uncertainty promptPotlapalli et al. (2023) | 28.71 | 0.891 |
| | Ours | 28.73 | 0.893 |
| CSUWB | w/o CUWB in CSUWB | 28.55 | 0.891 |
| | w/o SUWM in CSUWB | 28.59 | 0.891 |
| | w/ CSUWB←CBAMWoo et al. (2018) | 28.67 | 0.892 |
| | w/ CSUWB ← TransformerZamir et al. (2022) | 28.67 | 0.890 |
| | Ours | 28.69 | 0.894 |

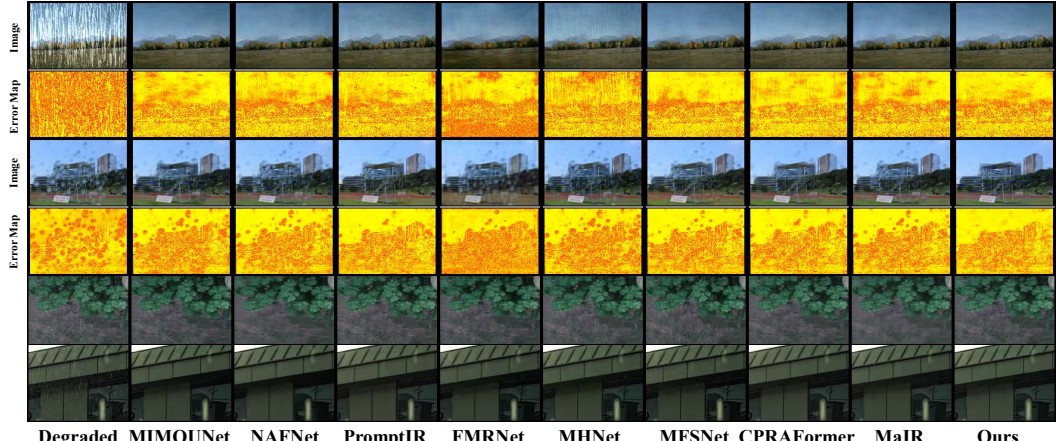

**Degraded  MIMOUNet  NAFNet  PromptIR  FMRNet  MHNet  MFSNet  CPRAFormer  MaIR  Ours**

Figure 5: Visual results of comparative methods and our UVPNet on synthetic rainy images and real-world rainy images.

Table 5: Comparative results on the assistance of segment priors and depth priors.

| Method | Venue | Rain100H | | Test100 | | Raindrop-B | |
|---|---|---|---|---|---|---|---|
| | | PSNR↑ | SSIM↑ | PSNR↑ | SSIM↑ | PSNR↑ | SSIM↑ |
| MFSNetGao & Dang (2025) | TCSVT'25 | 27.89 | 0.904 | 27.76 | 0.924 | 24.62 | 0.873 |
| MaIRLi et al. (2025) | CVPR'25 | 28.49 | 0.905 | 29.15 | 0.932 | 24.65 | 0.875 |
| 0R&0D | - | 28.60 | 0.912 | 29.07 | 0.936 | 25.22 | 0.890 |
| wR&wD | - | 28.69 | 0.913 | 29.21 | 0.936 | 25.28 | 0.892 |

time, UVPNet combines the segment priors provided by SAM and depth priors provided by Depth Anything V2 with distribution modeling, effectively extracting the visual priors in an uncertainty-aware manner. Furthermore, with the integration of certainty fusing uncertainty prompt learning, UVPNet excellently handles the model-wise epistemic uncertainty in the high-uncertainty rain streak and rain drop removal task. Comparative experiments on both publicly synthesized datasets for rain streak and rain drop removal and real-world rainy images, along with computing complexity and ablation studies, demonstrate the effectiveness and rationality of our method with and without the assistance of vision large models.

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

## A    APPENDIX

## B    THE USE OF LARGE LANGUAGE MODELS (LLMS)

The author(s) acknowledge that LLMs have been used to polish the language of this paper.

