# OpenReview forum: "RUSID: Robust Uncertainty-aware Single Image Deraining beyond Certainty"
_ICLR.cc/2026/Conference — Submitted to ICLR 2026_

### Official Review · Reviewer_RjkL · 2025-10-26

**Soundness:** 2
**Presentation:** 2
**Contribution:** 2
**Rating:** 4
**Confidence:** 5

**Summary:**

This paper introduces UVPNet, a single image deraining framework that addresses both aleatoric and epistemic uncertainties by integrating visual-prior-based probabilistic modeling and prompt interaction. The proposed framework consists of three key components: a Distribution-aware Visual Priors Learning (DVPL) module for data-wise uncertainty, a Certainty-Uncertainty Prompt Fusion (CUPF) module for epistemic uncertainty, and a Channel Spatial Uncertainty Weighting Block (CSUWB) for uncertainty-enhanced feature refinement.

**Strengths:**

1. The proposed approach is the first to integrate uncertainty modeling with visual priors in single image deraining, offering a new perspective for handling complex degradations.
2. The DVPL effectively captures aleatoric uncertainty using semantic and depth priors, while the CUPF fuses certainty and uncertainty prompts to enhance epistemic robustness, making the framework interpretable and principled.
3. The method achieves state-of-the-art performance across multiple synthetic and real-world benchmarks.

**Weaknesses:**

1. The paper lacks a deeper theoretical justification or quantitative insight into how uncertainty modeling directly contributes to the observed performance gains.
2. The paper suffers from an incomplete structure due to the absence of a dedicated and systematically organized “Related Work” section. Relevant prior studies are briefly mentioned in the introduction and method sections, but there is no comprehensive review of existing approaches in uncertainty modeling, deraining networks, or visual priors.
3. Although the model claims low complexity, the paper does not report training time or memory consumption, making the efficiency claims difficult to validate.
4. Comparisons do not include recent foundation or diffusion-based restoration methods, which may undermine the strength of the claimed superiority.
5. The paper provides insufficient visual comparisons, with a lack of qualitative examples against multiple state-of-the-art baselines. This omission prevents readers from visually assessing the improvements in detail preservation, artifact suppression, and perceptual quality.

**Questions:**

See the above parts.

---

### Official Review · Reviewer_hK5F · 2025-10-27

**Soundness:** 3
**Presentation:** 3
**Contribution:** 3
**Rating:** 4
**Confidence:** 5

**Summary:**

An uncertainty-aware work on image deraining with visual priors

**Strengths:**

1. This work explicitly models both aleatoric (data) and epistemic (model) uncertainty for single image deraining.

2. The authors provide extensive experiments that are convincing and thorough.

3. The ablation studies in Table 4 are detailed and effectively prove the contribution of each proposed component.

4. The paper is generally well-structured, with clear explanations of the problem, methodology, and results.

**Weaknesses:**

1. The paper does not specify how the certainty and uncertainty prompts are initialized. Are they randomly initialized, or are they derived from the input or features?

2. In Eq. (8), the output  is defined using PGM and PIM, but these modules are only cited and not described in the main text, leaving their specific function a bit vague for a reader not intimately familiar with PromptIR.

3. While the method performs well on the tested real-world dataset, a broader discussion of its generalization ability to more diverse and challenging real-world conditions (e.g., heavy rain, night rain, combined rain and fog) would strengthen the work. Why did the authors not to choose real-world benchmarks to conduct experiments?

4. This approach seems to put the cart before the horse, since the purpose of image deraining is precisely to improve the performance of high-level vision algorithms. Yet now we are using high-level visual information to guide the deraining process. Is this setup truly reasonable?

5. When utilizing two "anything" models, does the author's proposed method still maintain an efficiency advantage compared to current existing methods?

6. The authors need to add an overall pipeline diagram to illustrate the information flow of the entire method.

7. In fact, I feel this work is semi-finished and requires substantial improvements in motivation, methodology, experiments, and discussion.

**Questions:**

1. The authors are strongly advised to provide a clearer explanation of the motivation and methodology, as the current version appears quite confusing.

2. Why not conduct comparisons on real-world datasets, given the abundance of existing real rainy image datasets available?

3. Why is there inconsistency in the style of the figures?

4. I did not find the supplementary materials. In my opinion, the current work does not sufficiently demonstrate the significance to the research community.

---

### Official Review · Reviewer_ZMHb · 2025-10-31

**Soundness:** 2
**Presentation:** 2
**Contribution:** 2
**Rating:** 2
**Confidence:** 4

**Summary:**

This paper presents an uncertainty-based approach for robust single image deraining. The method incorporates uncertainty modeling into the Segment Anything Model (SAM) and depth priors. These priors, re-parameterized as Gaussian distributions, are injected into the deraining model as visual prompts. Furthermore, the approach introduces channel-wise and spatial-wise uncertainty into the deraining process, enabling variable deraining results from a fixed rainy input. The proposed approach has outperformed current SOTA methods on Rain13K and real-world rainy images.

**Strengths:**

- This paper introduces uncertainty into both the visual prompts and deep features
- The quantitive results demonstrate its superiority over the listed methods
- The proposed method presents better real-world deraining results compared to the listed methods.

**Weaknesses:**

- The paper omits a systematic literature review, particularly regarding uncertainty-based methods in image deraining and restoration. This makes it difficult to situate the proposed contributions within the existing research literature.
- While introducing uncertainty into visual prompts and deep features is a core contribution, the paper lacks critical supporting evidence. For instance, visualizations of the uncertainty maps are absent, leaving the reader to speculate on their form and function.
-  Quantitative comparison against other uncertainty-based methods is absent.
- Qualitative comparison in Figure 5 is not enough, with comparisons on only two synthetic and one real-world images.
- The analysis of the learned prompts is lacking. Incorporating visualization techniques (e.g., t-SNE, as used in PromptIR) would greatly help in interpreting what the uncertain prompts has learned.
- A significant discrepancy exists in the quantitative results. For example, on the Rain100L dataset, the paper reports 32.73 for CPRAformer, whereas the official paper reports 35.98. The authors must clarify their evaluation protocol in detail (e.g., Y channel vs. RGB).
- Organizational and Notational Inconsistencies (some of them are listed below):
    *   The training loss function is not presented in the paper.
    *   There are inconsistent notations, such as the use of $\odot$ in Eq. (6) versus $*$ in Eq. (13) for what appears to be the same operation.
    *   Figure 2 incorrectly uses "GAP" where it should be "GMP."
    *   Table 5 is missing the 0S and wS configurations discussed in Section 3.3.
    *   The venue "Nips'23" in Table 5 should be formatted correctly as "NeurIPS '23."

**Questions:**

- Given the proposed uncertainty modeling, what is the method's capability for training on datasets with various degradations (e.g., combining rain streaks, raindrops, and haze)? Does the uncertainty framework help the model handle these different degradations?
- Which component—the certain or the uncertain prompts—is primarily responsible for learning degradation-aware information?
- To further demonstrate practical utility, could the model be trained and evaluated on real-world datasets like the SPADataset? Providing more qualitative results on challenging real-world scenarios would strongly validate the method's robustness.

---

> ### Comment · Reviewer_ZMHb · 2025-11-28
> **Reviewer response**
>
> Given that the authors have not addressed the above concerns, I will maintain my score. However, I am open to raising the rating if the above issues could be addressed.

---

### Official Review · Reviewer_V41S · 2025-10-31

**Soundness:** 2
**Presentation:** 2
**Contribution:** 2
**Rating:** 4
**Confidence:** 4

**Summary:**

This paper introduces RUSID (UVPNet), a single-image deraining network that aims to handle both data and model uncertainties using visual priors and prompt fusion. It includes three main components: DVPL for data uncertainty, CUPF for model uncertainty, and CSUWB for feature reweighting. Experiments on synthetic and real rainy datasets show small improvements over recent baselines.

**Strengths:**

(1) The paper is clearly organized and supported with detailed architecture figures and ablation studies.

(2) Combining uncertainty modeling and prompt-based priors is an interesting direction that could inspire further work.

(3) The experimental coverage across different datasets and metrics is relatively complete.

**Weaknesses:**

(1) The method piles together many fashionable ideas like uncertainty, prompts, and visual priors, but the technical substance is shallow. Each proposed block is only a small variant of existing attention or distribution modules with new names.

(2) The “uncertainty modeling” is not really probabilistic. Using a Softplus-activated variance map does not provide genuine uncertainty estimation or calibration.

(3) There is no clear evidence that modeling uncertainty actually helps deraining. The paper lacks analysis, visualization, or any metric showing that uncertainty estimates correlate with better restoration.

(4) The reliance on SAM and Depth Anything priors raises questions about practicality. These huge models make the pipeline heavy, and the claimed efficiency is not credible.

**Questions:**

(1) What concrete benefit does the uncertainty modeling provide? Can you show calibration curves or any statistical validation?

(2) How is CUPF fundamentally different from the prompt fusion in PromptIR? The difference seems marginal.

(3) The ablation shows very little performance change when SAM and Depth Anything priors are removed. Does that not undermine the main motivation?

(4) How can this approach be deployed in real applications given its dependency on large pretrained priors and high memory cost?

---

> ### Author Response · Authors · 2025-11-22
> **Reply to Reviewer V41S**
>
> Thank you very much for your constructive feedback. I will make revisions based on your suggestions. Especially your opinion 'Can you show calibration curves or any statistical validation?', I think this opinion is very helpful in improving the quality of the paper, but I don't quite understand the specific meaning or meaning of 'calibration curves' and' statistical validation 'you mentioned. Please explain it specifically.

---

### Meta-Review · Area_Chair_K7HL · 2026-01-01

**Summary:**

The paper introduces a framework called the Uncertainty-aware Visual-priors Prompt-interaction Network (UVPNet) for  single image deraining.  The reviewers concerns that the method superficially combines several fashionable concepts without substantial technical depth; there is no clear evidence that modeling uncertainty genuinely aids deraining; it lacks a systematic literature review, especially concerning uncertainty-based methods in image deraining and restoration; quantitative comparisons with other uncertainty-based approaches are absent; and it fails to specify how the certainty and uncertainty prompts are initialized.

Considering all reviewers' evaluations and scores, I believe this paper does not yet meet the publication standards of ICLR.

**Reviewer Concerns:**

No concern is addressed.
The following concerns are still outstanding:
1) The method piles together many fashionable ideas, but the technical substance is shallow.
2) There is no clear evidence that modeling uncertainty actually helps deraining.
3) The paper omits a systematic literature review, particularly regarding uncertainty-based methods in image deraining and restoration.
4) Quantitative comparison against other uncertainty-based methods is absent.
5) The paper does not specify how the certainty and uncertainty prompts are initialized.

**Reviewer Scores:**

Since the authors did not respond to the reviewers' questions, the reviewers' scores will remain unchanged.

---

### Decision · Program_Chairs · 2026-01-26

Reject